# Imaging of Pediatric Testicular and Para-Testicular Tumors: A Pictural Review

**DOI:** 10.3390/cancers14133180

**Published:** 2022-06-29

**Authors:** Anne-Laure Hermann, Aurore L’Herminé-Coulomb, Sabine Irtan, Georges Audry, Liesbeth Cardoen, Hervé J. Brisse, Saskia Vande Perre, Hubert Ducou Le Pointe

**Affiliations:** 1Department of Pediatric and Prenatal Imaging, Armand-Trousseau Hospital, APHP, Sorbonne University, 75012 Paris, France; saskia.vandeperre@aphp.fr (S.V.P.); hubert.ducou-le-pointe@aphp.fr (H.D.L.P.); 2Department of Pathology, Armand-Trousseau Hospital, APHP, Sorbonne University, 75012 Paris, France; aurore.coulomb@aphp.fr; 3Department of Pediatric Surgery, Armand-Trousseau Hospital, APHP, Sorbonne University, 75012 Paris, France; sabine.irtan@aphp.fr; 4Department of Imaging, Institut Curie, 75005 Paris, France; georges.audry@aphp.fr (G.A.); liesbeth.cardoen@curie.fr (L.C.); herve.brisse@curie.fr (H.J.B.)

**Keywords:** pediatric testicular tumors, pre-pubertal tumors, germ cell tumors, non-germ cell tumors, imaging, ultrasound, color Doppler, scrotal MRI

## Abstract

**Simple Summary:**

Pediatric testicular tumors are rare and represent 1% to 2% of all solid tumors in children. Germ-cell tumors are the most frequent etiology; most are benign in the pre-pubertal population, whereas post-pubertal testicular tumors are similar to those that occur in adults, with potential malignancy. The development of testis-sparing surgery for presumed benign lesions enhances the need for better pre-operative tumor characterization by imaging, among other tools. Ultrasonography (US) remains the first-line examination for scrotal pathology assessment, but other techniques such as contrast-enhanced US, elastography, or scrotal MRI are emerging modalities for scrotal evaluation both in children and adults. This review provides an overview of imaging features of the most frequent testicular and para-testicular tumor types in children and discusses the role of imaging in disease staging and monitoring children with testicular tumors or risk factors for testicular tumors.

**Abstract:**

Pre- and post-pubertal testicular tumors are two distinct entities in terms of epidemiology, diagnosis and treatment. Most pre-pubertal tumors are benign; the most frequent are teratomas, and the most common malignant tumors are yolk-sac tumors. Post-pubertal tumors are similar to those found in adults and are more likely to be malignant. Imaging plays a pivotal role in the diagnosis, staging and follow-up. The appearance on ultrasonography (US) is especially helpful to differentiate benign lesions that could be candidates for testis-sparing surgery from malignant ones that require radical orchidectomy. Some specific imaging patterns are described for benign lesions: epidermoid cysts, mature cystic teratomas and Leydig-cell tumors. Benign tumors tend to be well-circumscribed, with decreased Doppler flow on US, but malignancy should be suspected when US shows an inhomogeneous, not-well-described lesion with internal blood flow. Imaging features should always be interpreted in combination with clinical and biological data including serum levels of tumor markers and even intra-operative frozen sections in case of conservative surgery to raise any concerns of malignity. This review provides an overview of imaging features of the most frequent testicular and para-testicular tumor types in children and the value of imaging in disease staging and monitoring children with testicular tumors or risk factors for testicular tumors.

## 1. Introduction

Imaging plays different roles in the management of testicular tumors in children and adolescents. First and principally, it could help advance the diagnosis. Ultrasonography (US) and MRI are the two imaging modalities used to evaluate testicular pathology. Importantly, imaging can distinguish intra-testicular and para-testicular locations related to different diagnostic ranges including para-testicular rhabdomyosarcoma, the most common malignant tumor of the genito-urinary tract in the first two decades of life. For intra-testicular lesions, imaging plays a major role in differentiating benign and malignant testicular neoplasms, which are treated differently. For decades, testis-sparing surgery (TSS) preserving testicular function has become a well-accepted approach for benign testicular tumors in pre-pubertal patients, whereas malignant lesions are treated by radical orchidectomy [1]. 

With malignancy, imaging plays a role in the staging of the tumor. Chest radiography and abdominal US or chest, abdominal and pelvic CT scans should be used to accurately stage malignant tumors, with particular attention to depicting retroperitoneal lymph nodes or pulmonary metastases [2]. 

Imaging also plays an important role in the post-operative follow-up, especially to depict recurrence in the contralateral testicle or in distant sites in case of malignant tumors treated with orchidectomy. It also helps to evaluate the volume of residual normal testicular tissue after TSS which could be an indicator of fertility [3]. 

Finally, imaging plays a role in screening precancerous lesions that could develop in patients with risk factors for the development of testicular tumors. 

This review is based on a selection of recent papers mainly from 2010 to 2022 in the *PubMed* database dealing principally with imaging of intratesticular tumors in a pediatric population. We used keywords for each imaging modality or for each tumoral histotype. Papers concerning imaging for main para-testicular tumors as differential diagnoses or about testicular microlithiasis management in children were also reviewed.

This review provides an overview of the value of multiparametric US and MRI in the diagnosis, staging and monitoring of testicular and a few cases of para-testicular tumors in children and adolescents. We briefly address the relationship between testicular microlithiasis and tumor development. The main objective was to review the imaging features of various histological subtypes of pediatric testicular tumors to provide diagnostic criteria in favor of a benign diagnosis that could lead to a conservative therapeutic strategy.

## 2. Epidemiology of Pediatric Testicular Tumors

Pediatric testicular tumors are rare and represent 1% to 2% of all pediatric solid tumors. In patients < 14 years old, the incidence is estimated at 0.5 to 2 per 100,000 children [4,5]. Testicular tumors have a bimodal distribution: the first peak is seen in boys < 3 years of age, most being benign, and the second occurs after puberty, from age 15 to 19 years, with a higher proportion of malignant neoplasms. This observation highlights two distinct patterns of disease leading to different approaches to treatment. 

Among all prepubertal intra-testicular tumors, 90% are germ-cell tumors (GCTs) and 10% to 15% are epidermoid cysts [6,7,8]. Reports suggest that benign GCTs are frequent, approximately 40% to 50% being teratomas. Most teratomas show mature elements, but immature teratomas in this age group have been reported [9]. Epidermoid cysts are always benign. Malignant GCTs are mainly represented by yolk-sac tumors, with a frequency ranging from 8% to 30% [7]. Among non-GCTs, Leydig cell tumors are more frequent, representing approximately 5% of the etiologies, followed by juvenile stromal granulosa cell tumors. The incidence of prepubertal testicular tumors peaks in the first three years of life [10]. As compared with yolk-sac tumors, teratomas usually develop in younger children, with a median age of approximately 8 to 13 months and 16 to 17 months, respectively [3,7]. Among prepubertal testicular malignancies, the 5-year relative survival is 97% for children with localized tumors and 73% for those with distant disease [11]. 

Between ages 4 to 11 years, the diagnosis of a testicular GCT is rare. In contrast, in post-pubertal adolescents 15 to 19 years old, the incidence increases and represents 12% of all cancers in adolescents [12]. In this population, tumors are more likely to be malignant and include immature teratomas, mixed non-seminomatous malignant GCTs and embryonal carcinomas [13]. In young adults, 95% of testicular cancers are malignant GCTs, and 50% of these are seminomas. The 2016 World Health Organization classification system redefines testicular GCTs into two main groups: those derived from germ-cell neoplasia in situ, which are representative of post-pubertal lesions, and those not derived from germ-cell neoplasia in situ, which characterize prepubertal neoplasms [14]. 

One specific tumor type is the gonadoblastoma, which includes both germ-cell and sex-cord stromal-cell tumor types and occurs almost exclusively in the setting of differences in sexual differentiation [15]. In patients presenting a scrotal mass, a para-testicular mass should also be accounted for in the differential diagnosis. Para-testicular rhabdomyosarcoma is the most frequent para-testicular malignant mass in children; it represents 5% of all testicular and para-testicular malignancies. The para-testicular tumor spectrum remains wide and also includes benign lesions such as hemangiomas, leiomyomas, fibromas, lipoblastomas [16], cystic lymphangiomas and the rare malignant melanotic neuroectodermal tumor of infancy [17]. A classification of testicular tumors in children is presented in Table 1. 

Testicular tumor markers are a major tool in evaluating pediatric testicular tumors; these include alpha-fetoprotein (AFP) and beta-human chorionic gonadotrophin (BHCG). Serum AFP levels are normally higher in children than in adults and are reduced to normal levels by age 1 (<10 ng/mL). Therefore, for children < 1 year old with testicular tumors, AFP level may be elevated in those with benign tumors, whereas for children > 1 year, a normal AFP level often indicates a benign tumor. Elevated serum AFP level is strongly associated with >90% of yolk-sac tumors, whereas immature teratomas may present a slightly elevated serum AFP level [18]. AFP should be measured before any therapeutic intervention and after surgery to assess an appropriate decrease in level. 

BHCG is elevated in choriocarcinomas, embryonal carcinomas or seminomas, which are extremely rare in prepubertal boys but could be seen in adolescents. Thus, it is not useful in the diagnostic work-up of prepubertal boys with a testicular tumor but is useful in adolescents presenting a testicular mass [3]. Lactate dehydrogenase is also useful because high levels are associated with bulky disease and increasing levels after therapy may signify disease recurrence. 

## 3. Management of Pediatric Testicular Tumors

For decades, TSS has been reported to control pediatric testicular benign tumors, with potential psychologic, cosmetic and functional advantages. The size of the tumor is an important factor in choosing the therapeutic option, with significantly smaller lesions in children after TSS versus radical orchidectomy [4]. Actually, a cut-off tumor size of <2.5 cm is generally considered a good candidate for TSS [4]. TSS has shown reliability and feasibility with no atrophy and rare cases of recurrence [1,19]. A recent systematic review [20] approved this strategy in selected cases of prepubertal testicular tumors. 

However, tumor size only does not seem a sufficient criterion to differentiate benign from malignant neoplasms. Imaging, especially US, when combined with clinical data and tumor marker levels, is a reliable protocol for the differential diagnosis of benign and malignant lesions. However, intra-operative frozen section analysis seems to more accurately predict the pathological diagnosis [21], which suggests that it should be systematically performed during TSS to confirm the benign diagnosis and that microscopically margin-negative resection has been achieved. In the case of microscopic positive margins in a malignant or potentially malignant tumor, orchidectomy should be performed [3]. Orchidectomy with the inguinal approach should be considered if normal testicular parenchyma is no longer detectable on preoperatively high-resolution US and/or with AFP level > 100 ng/mL in a child > 12 months, considering high suspicion of yolk-sac tumor [3], or with AFP level > 23 ng/mL in a child < 8 months with a tumor > 2.5 cm because of increased frequency of immature teratomas [22]. 

## 4. Role of Imaging for Pediatric Intra-Testicular Tumor Characterization

### 4.1. Imaging Modalities

A high-resolution US (7.5–12.5 MHz) study based on gray-scale and color Doppler is the standard initial test for assessing testicular neoplasms because of its low cost, wide availability, and high sensitivity. In children, US is particularly advantageous because it is safe and radiation-free and with no need for sedation or general anesthesia during the imaging evaluation.

Contrast-enhanced US (CEUS) with Sonazoid, a blood pool-contrast agent (consisting of microbubbles of perfluorobutane gas coated with hydrogenated egg phosphatidylserine sodium with a mean size of 2.3–2.9 μm), has been used in adults with small lesions to characterize local microvascular dynamics. In adults, a few case reports and limited series have been published, with discordant results [23,24]. In children, scrotal CEUS can confidently establish testicular tissue vascularity even in the small-volume pediatric testes, but it is performed in an off-label pattern. CEUS may be considered appropriate when its benefits outweigh the potential risks associated with alternative imaging investigations, such as the need to use sedation or general anesthesia [25]. There are no standardized dosage schemes for the administered US-contrast agent for pediatric scrotal CEUS [25].

Elastography is classified as strain and shear-wave elastography (SWE). Strain elastography assesses the amount of target tissue deformation relative to surrounding parenchyma, and shear-wave elastography measures the speed of shear waves to obtain a quantitative estimation of tissue stiffness. This technique seems promising in adults but needs further investigation [26]. 

In children, Huang et al. promoted the use of multiparametric US, including simultaneous use of conventional modes with CEUS and tissue elastography, particularly when conventional US findings are equivocal in the evaluation of solid tumors [25]. The technique could help increase diagnostic confidence in favor of a malignant tumor when a lesion is hyper-enhanced on CEUS and “hard” on strain elastography. 

Scrotal MRI has been proposed as an additional imaging technique to US in cases of uncertainty. As in adults, in children, it can help in precisely localizing testicular tumors, tissue characterization and assessing the involvement of surrounding structures [27,28]. The recommended MRI protocol should include T1- and T2-weighted imaging, diffusion-weighted imaging and dynamic contrast-enhanced MRI [29]. In adults, diffusion-weighted MRI has shown its utility to differentiate malignant lesions from benign lesions with a cut-off ADC value of 0.9 × 10^−3^ mm^2^/s (sensitivity of 71%, specificity of 72%, and accuracy of 71%) [30] and conventional MRI may be discriminative to differentiate seminoma from non-seminomatous germ-cell tumors [31]. A combination of SWE elastography and diffusion-weighted MRI has also been evaluated in patients with normal testicular tissue, testicular microlithiasis or testicular cancer, with significantly higher values of elasticity and lower ADC values in patients with testicular cancer compared to patients with normal testicular tissue or microlithiasis [32]. This suggests that MRI diffusion and elastography may be useful to preoperatively differentiate benign from malignant testicular lesions, but this should be confirmed in children with testicular tumors.

### 4.2. Benign GCTs

#### Mature Teratomas and Dermoid Cysts

Histologically, teratomas are a combination of the three primitive embryological germ cell layers: ectoderm, mesoderm and endoderm. On sonography, most of the teratomas have mixed cystic and solid components, but they could also be mainly cystic or mainly solid lesions, with areas of calcification [22] (Figure 1). Mature cystic teratomas, also known as dermoid cysts, differ from epidermoid cysts because they contain skin and skin appendages, including hair follicles and sebaceous glands [33]. In rare cases, a cystic mature teratoma could be suspected prenatally before an intra-abdominal cyst in a male fetus with undescended homolateral testis [34]. For pre-pubertal males, TSS could be performed safely with a lack of evidence of malignancy pre-operatively and frozen sections raise no histological concerns about malignancy. 

### 4.3. Epidermoid Cysts

These are of ectodermal origin and seem to be related to well-differentiated teratomas. The classical sonographic pattern of an epidermoid cyst is a round well-demarcated avascular intra-testicular mass with a “target” or “onion-ring appearance” of multiple concentric layers of alternating echogenicity and an outer hyperechoic rim (Figure 2). Keratin-producing epithelium is responsible for the keratin accumulation within the lumen of the cyst that appears hyperechoic on US [22]. 

The sonographic appearance of teratomas and epidermoid cysts of the testis overlap. Although the onion-ring appearance of an intra-testicular tumor suggests an epidermoid cyst, this appearance may also be found in teratomas. CEUS could be used to demonstrate the absence of internal vascularity [25]. TSS is the standard management. 

### 4.4. Malignant GCTs

#### 4.4.1. Yolk-Sac Tumor 

Yolk sac tumors of the testes are the most common malignant GCTs in children, accounting for 70% to 80% of malignant testicular neoplasms. Children usually present asymptomatic testicular enlargement within the first two years of life (median age of 16 months) with a high percentage (80–85%) presenting early-stage disease [35]. An elevated serum AFP level is closely associated with yolk-sac tumors in more than 90% of children as the tumoral cells synthetize AFP. Elevated AFP levels could also be found in mixed GCTs containing a yolk-sac component. The prognosis of yolk-sac tumors depends on early detection and treatment. Although if non-specific, the sonographic appearance is frequently an ovoid, homogeneous, well-circumscribed, isoechoic, solid testicular mass with good sound-through transmission and increased internal and peripheral vascularity on color Doppler imaging, without any calcification [36,37] (Figure 3). However, an enlarged inhomogeneous testis in the absence of a mass, cystic foci or echogenic foci representing an area of hemorrhage has also been reported [35]. Macroscopically, tumors are mostly solid and yellow-grey to tan cut surface. In clinical practice, a sonogram depicting a solid testicular hypervascular mass in a child < 2 years old combined with an elevated AFP level (>100 ng/mL) could suggest the diagnosis [38]. Inguinal orchidectomy and postoperative chemotherapy is the standard care to prevent recurrence [37,39]. Roughly 20% of children with stage I disease (no metastatic disease on MRI of the abdomen and CT scan of the chest and normal post-operative age-adapted AFP level) may show metastases within the next 2 years. Therefore, serum AFP levels should be monitored and an MRI of the abdomen performed postoperatively every 2 to 3 months to gauge recurrence during follow-up at least for the first 2 to 3 years [3]. Lymphovascular invasion and necrosis were found to be independent risk factors for relapse [40]. However, there is still controversy concerning the use of retroperitoneal lymph-node dissection in patients at stage I of the disease. Some studies even suggest that chemotherapy could be omitted for stage I patients < 1 year old with AFP level stabilizing at a normal level [37]. The overall 4-year survival rate remains excellent, from 95% to 98% [40].

#### 4.4.2. Immature Teratomas

Sonographic appearance is not specific enough to differentiate mature and immature teratomas or benign and malignant teratomas because immature/malignant teratomas also show mixed components with calcification [22]. The criteria favoring immature teratomas in prepubertal patients are the young age of the child, a larger tumor size and increased AFP level, with a cut-off of 8 months, 2.5 cm and 23 ng/mL, respectively [22]. These criteria could be considered for orchidectomy rather than TSS. Analysis of per-operative frozen sections is also important to approach the diagnosis of a benign or malignant tumor and to ensure complete removal. It can also assess the presence or not of pubertal changes in the adjacent testicular parenchyma. Indeed, a teratoma with pubertal changes is considered malignant. In these cases, radical orchidectomy and surveillance protocols similar to those in adults should be applied. In the post-pubertal population, teratomas are rarely found in pure form but are associated in approximately half of the cases to mixed non-seminomatous malignant GCTs [39].

#### 4.4.3. Mixed Non-Seminomatous Malignant GCTs

In post-pubertal patients, testicular tumors develop from intratubular germ-cell neoplasia. The increased testosterone level associated with puberty changes stimulates dormant germ cells to divide along two malignant pathways. One distinct pathway leads to non-seminomatous histology: relatively undifferentiated embryonal carcinoma, teratoma, trophoblastic choriocarcinoma, and yolk-sac tumor; the other pathway leads to seminoma [39]. Embryonal carcinoma is the most common malignant GCT in adolescents, frequently presenting with aggressive behavior. It could produce BHCG and also AFP. Choriocarcinoma is the rarest tumor but most aggressive GCT, with a high frequency of lung or brain metastases and elevated BHCG levels. On imaging, a typical pattern of malignancy is an inhomogeneous, not-well-described intra-testicular mass containing solid and cystic components, with increased internal blood flow [41] (Figure 4). Surgical excision of the tumor is the first step in treatment. Even disseminated disease is treatable with chemotherapy, retroperitoneal lymph-node dissection and targeted treatment. However, adolescents remain [39,42] with the highest mortality with a 5-year survival rate of 70% in case of metastatic disease.

#### 4.4.4. Seminomas

Although seminomas are the most common type of testicular GCT in adults, they are exceedingly rare in the pediatric and adolescent populations. Hence, we do not discuss the imaging features of seminoma here. 

### 4.5. Non-GCTs: Sex-Cord Stromal Tumors

#### 4.5.1. Leydig Cell Tumors (LCTs)

LCTs are rare tumors representing 0.4% to 9% of all testicular tumors in pre-pubertal children. Precocious puberty due to excess testosterone production is frequently associated and results from Leydig cell hyperplasia secondary to activating mutations in the luteinizing hormone receptor [43]. LCT should be suspected in boys with a testicular mass and early-onset puberty along with high testosterone and low gonadotrophin levels. The precocious puberty could be limited to the ipsilateral hemi-scrotum [8]. Children are usually 5 to 10 years old. On US, typical LCTs appear as an isolated moderately hypoechoic mass with well-defined margins without microlithiasis and predominant hypervascularization [44]. In children, some additional features have been described such as the central vascularization on ultrasensitive Doppler or the hyperechoic rim surrounding the lesion, that could be useful to distinguish LCT from testicular adrenal rest tumors (Figure 5) [45]. On CEUS with Sonazoid, LCTs demonstrate a strong and fast enhancement pattern followed by a delayed washout in the lesion [46]. On elastography, case series report a moderate increase in stiffness [26,45]. On MRI, LCTs have been reported with low signal intensity on T2-weighted imaging, with enhancement after contrast administration, which could mimic the aspect of malignant testicular tumors. However, in some studies, dynamic contrast-enhanced MRI could help differentiate LCTs from malignant neoplasms, with LCTs displaying a pattern of rapid and marked wash-in followed by prolonged wash-out [47]. High vascularization in LCTs could be explained by an intense expression of endocrine gland-derived vascular growth factor, whereas delayed wash-out is due to less arteriovenous fistula than in malignant lesions [48]. This aspect seems characteristic for LCTs but remains non-specific in the adult literature [49]. Macroscopically, LCTs appear as yellow-brown nodules [50]. Because most LCTs are benign small lesions with no recurrence or malignant behavior documented in the pediatric population, TSS has been advocated as an appropriate therapeutic procedure for these tumors [51,52]. After surgery, endocrine long-term follow-up is required because androgenic effects and precocious puberty are not reversed after removal [50].

#### 4.5.2. Sertoli-Cell Tumors

Sertoli cell tumors account for <1% of testicular tumors and can occur in children in 20% of cases, usually within the first year of life. In the pediatric population, large-cell calcifying Sertoli cell tumors are the most common tumor variant [53]. They are sporadic in 60% of the reported cases but may be associated with the Carney complex [54] or Peutz–Jeghers syndrome [55]. Except for one case of malignant large-cell calcifying Sertoli tumors described in a child [53], they represent mostly a benign entity in the pediatric population; therefore, TSS should be performed. Due to the frequent higher expression of aromatase within these tumors, there is higher conversion of testosterone to estradiol. Therefore, Sertoli-cell tumors in a prepubertal boy may present by acceleration of his growth and gynecomastia. On sonography, tumors are characterized by partially calcified, regularly round hyperechoic intra-testicular lesions with acoustic shadowing and vascularity. Internal vascularization could be difficult to assess because of artifacts caused by prominent calcifications (Figure 6) [56]. 

#### 4.5.3. Juvenile Granulosa Cell Tumors

These are rare sex-cord stromal tumors accounting for 1% to 5% of all prepubertal testis tumors. Such tumors usually occur in the first year of life, within the first 6 months. A few cases of prenatal diagnosis have been reported [57,58]. The most common presentation is a painless testicular mass or an enlargement of the testis with possible hormonal changes including gynecomastia and anomalies of puberty [59]. The tumors have a wide morphologic spectrum with no specific imaging appearance described. They are classically well-circumscribed, with a median size of 1.5 cm, and could contain cystic components or be entirely solid [59] (Figure 7). Macroscopically, they have a typical yellow-tan appearance. The solid and reticular patterns may pose diagnostic pitfalls, but the lobular appearance and the follicular differentiation are characteristic of the histology. Immunohistochemically patterns could aid in their distinction from other tumors of young boys, particularly yolk-sac tumors, which usually peak at a later age [59]. The juvenile type represents a benign entity, with low recurrence described so far after TSS [60]. In contrast, adult testicular granulosa cell tumors have metastatic potential [60]. 

### 4.6. Lymphoma and Leukemia

Testicular infiltration is seen in up to 8% of children with leukemia, and malignant lymphoma represents 1% to 7% of all testicular tumors. These diagnoses should be considered in children presenting a painless testicular enlargement [61]. Gray-scale sonograms of lymphoma and leukemia typically show unilateral or bilateral enlargement of the testis with poorly defined diffuse or focal regions of decreased echogenicity and maintenance of the normal ovoid testicular shape. Color Doppler imaging is essential, demonstrating increased intralesional flow within all areas of leukemic or lymphomatous involvement relative to normal testicular parenchyma, with preservation of normal vascular architecture [62] regardless of the size of the infiltration [61] (Figure 8). These features could mimic the appearance of an inflammatory process of the testis, but the lack of pain, epididymal enlargement or hyperemia and thickening of scrotal skin are more suggestive of a neoplastic process than orchi-epididymitis. 

### 4.7. Intra-Testicular Metastases of Primary Solid Tumor

Secondary testicular involvement of a primary tumor is extremely rare in children. Some cases of testicular involvement have been described in metastatic neuroblastoma, the most common extracranial solid tumor of childhood. Testes may be sanctuary sites when neuroblastoma is metastatic (Figure 9), as in leukemia [63].

### 4.8. Other Rare Diagnoses of Intra-Testicular Tumors

Some other rare diagnoses of intra-testicular pediatric neoplasms reported include myeloid sarcoma [64] or benign lesions such as myofibromas [65], intra-testicular hemangiomas [66], papillary cyst adenomas [13] or simple intra-testicular cysts [5].

### 4.9. Testicular Adrenal Rest Tumors

Testicular adrenal rest tumors are benign tumors found in the testes of males with congenital adrenal hyperplasia, an autosomal recessive disorder with an incidence of 1 to 20,000 live births [67]. Testicular adrenal rest tumors are often bilateral, located at the hilum of the testis and could be responsible for reduced fertility [68]. The imaging pattern on US is important to recognize in a pediatric population in order to consider the diagnosis of congenital adrenal hyperplasia and to differentiate them from other testicular tumors. The typical aspect on US is bilateral, mostly hypoechoic or hyperechoic lesions located at the hilum of the testis showing a multilocular pattern with no mass effect and normal testicular vessels coursing through the lesions (Figure 10) [69]. Their vascularization has also been reported as poor in recent case series [70]. CEUS and strain elastography could be performed, showing increased vascularization and increased stiffness in the hypoechoic areas [71]. MRI usually shows bilateral lobulated, well-demarcated, intra-testicular lesions that appear predominantly isointense or slightly hyperintense on T1-weighted imaging and hypointense on T2-weighted imaging, with heterogeneous and marked enhancement after gadolinium administration as compared with normal parenchyma [69,71].

## 5. Role of Imaging in Pediatric Extra-Testicular Tumor Characterization

Solid or complex extra-testicular scrotal masses, which can mimic intra-testicular tumors, represent a range of diagnoses commonly encountered in the pediatric population. These include congenital conditions and various benign lesions (Figure 11 and Figure 12) but also life-threatening malignancies such as rhabdomyosarcoma. 

Para-testicular rhabdomyosarcoma represents 7% of all rhabdomyosarcomas, the most common soft-tissue tumor in children. The distribution is bimodal, with peak incidence at 5 and 16 years and median age at diagnosis of 7 years. Rhabdomyosarcoma is a highly aggressive tumor, with frequent cases of regional lymph-node metastases at diagnosis. The typical presentation is a short history of a painless unilateral lump in the scrotum. Embryonal rhabdomyosarcomas represent 90% of all cases of para-testicular rhabdomyosarcomas in children. US commonly shows a large, ill-defined, heterogeneous vascular mass (Figure 13 and Figure 14). MRI could also be useful in cases of diagnostic uncertainty to reduce the risk of delayed diagnosis. MRI usually shows a solid para-testicular mass with heterogeneous T2-weighted signals and homogenous post-contrast enhancement [72]. 

## 6. Role of Imaging in Staging and Monitoring Pediatric Testicular Tumors

### 6.1. Staging of Pediatric Testicular Tumors and Para-Testicular Rhabdomyosarcoma

The chest and abdomen should be explored for malignant tumor staging and identification of metastases, the main sites being lungs and retroperitoneal lymph nodes. 

In testicular GCTs, abdomino-pelvic CT is routinely recommended for evaluation of the retroperitoneal lymph node stage and abdominal MRI provides results equivalent to a CT scan. A chest CT scan is the most sensitive examination for the detection of pulmonary metastases or mediastinal adenopathy. In routine practice, a thoraco-abdomino-pelvic CT scan is performed. Brain imaging is recommended in case of neurological symptoms or in subjects at risk for brain extension. ^18^F flurodesoxyglucose positron emission tomography (^18^F FDG PET) is not recommended in the initial staging of GCTs [73]. In pediatric and adolescent rhabdomyosarcomas, ^18^F FDG PET/CT or positron emission tomography/MRI along with chest CT scan are recommended for detecting and evaluating loco-regional and distant metastatic disease [74].

### 6.2. Monitoring Children with Testicular Tumors

Regular testicular US is recommended in the follow-up to detect any recurrence that could occur even after partial surgery of benign lesions and reported after excision of epidermoid cysts [8]. 

Despite no clear recommendations concerning the interval and duration of follow-up, the European Association of Urology-European Society of Pediatric Urology guidelines advise US examination every 3 to 6 months within the first year for benign lesions and beyond the first year after surgery for malignant tumors [3]. Long-term follow-up is not necessary for benign tumors. 

The rate of atrophy after TSS is extremely rare, and even reconstitution of a normal testicular form has been described after conservative surgery of a large intra-testicular cyst with little visible parenchyma [8]. 

### 6.3. Monitoring Children with Risk Factors for Testicular Tumors

Risk factors for post-pubertal testicular GCT are cryptorchidism, family history and infertility. Undescended testes are associated with increased risk if orchiopexy is performed after puberty [75]. The risk is increased three-fold with a family history of testicular GCT in a first-degree relative [76]. In contrast, the clinical significance of testicular microlithiasis remains unclear. In testicular microlithiasis, punctate calcifications are present in at least one testicle. Microlithiasis can be focal, multifocal or diffuse as well as unilateral or bilateral (Figure 15). The reported prevalence of microlithiasis in children ranges from 0.7% to 5.2% depending on the study [77,78,79]. Two retrospective studies [79,80] showed an association between microlithiasis and the development of primary testicular neoplasia, including malignant and benign GCTs or sex-cord stromal tumors. According to Trout et al. [79], the odds of any type of testicular neoplasia are 14-fold greater and that of malignant GCT 22-fold greater for boys with than without microlithiasis. However, consensus on monitoring strategies in these patients is lacking. A recent systematic review of the literature including the follow-up of 595 children [81] showed that only one child with testicular microlithiasis exhibited testicular neoplasia during puberty. This observation suggests that testicular microlithiasis in children may be a common incidental finding on US with no clear association with testicular malignancy. However, children with testicular microlithiasis and additional risk factors for testicular tumors, such as cryptorchidism, are encouraged to perform monthly self-examination from puberty. Still, there is no evidence that regular US examination is useful [82]. Interestingly, elastography has been evaluated in pediatric patients with microlithiasis in comparison to normal control subjects in the study of Bayramoglu et al. [83], demonstrating that elasticity values were significantly higher in testes with microlithiasis as compared with the control group. This suggests that SWE elastography should be a reliable method in follow-up examinations for pediatric testicular microlithiasis.

## 7. Conclusions

Pre- and post-pubertal testicular tumors are two distinct entities in terms of epidemiology, diagnosis and treatment. Most pre-pubertal tumors are benign, the most frequent being teratomas, and the most common malignant tumors are yolk-sac tumors, whereas post-pubertal tumors are similar to those found in adults and are likely more malignant. The long-term survival rates in children with testicular tumors are globally excellent after appropriate treatment. Imaging, especially the US appearance is helpful to differentiate benign lesions that could be candidates for TSS from malignant ones that require radical orchidectomy. Benign tumors tend to be well-circumscribed, with decreased Doppler flow. However, malignancy should be suspected when US shows an inhomogeneous, not-well-described lesion with internal blood flow. Few specific imaging patterns are described, except for some benign lesions with the onion-ring appearance of epidermoid cysts, the pure cystic mass of mature cystic teratomas and the mass with a typical internal and peripheral rim pattern of LCTs associated with precocious puberty. MRI is recommended in case of inconclusive characteristics on US imaging. However, this review was based mainly on papers dealing with a small number of patients or case reports describing the imaging features of a few cases. In any case, imaging features should always be interpreted in combination with clinical data including the age of the child and signs of hormonal secretion, the tumor size and biological data including serum levels of tumor markers. Moreover, analysis of intra-operative frozen sections is recommended in case of organ-sparing surgery to raise any concerns of malignancy. This strategy is included in an algorithm proposed by Kooij et al. [20] and allows for successful TSS in most pre-pubertal patients with testicular tumors. Imaging also plays a role in the initial staging of malignant tumors and the follow-up to detect any recurrence, particularly in the case of malignant neoplasms.

## Figures and Tables

**Figure 1 cancers-14-03180-f001:**
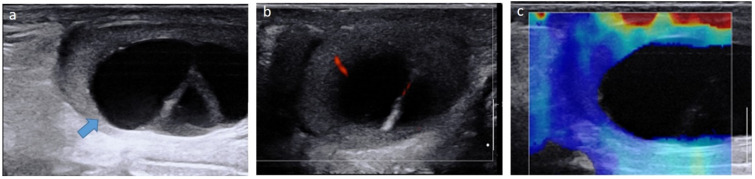
Mature cystic teratoma in a 4-month-old boy. Longitudinal gray-scale B mode (**a**), color Doppler (**b**) and strain elastography (**c**) US of the testis. Images show a cystic intra-testicular lesion (blue arrow) containing two inner septa with color Doppler signals and no signals on elastography. Serum levels of tumor markers were normal. The patient underwent conservative surgery. US, ultrasonography.

**Figure 2 cancers-14-03180-f002:**
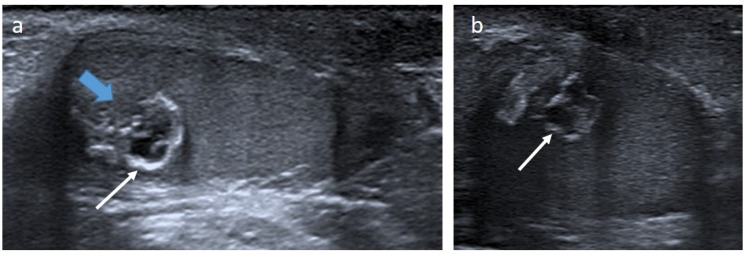
Epidermoid cyst in a 20-month-old boy. Longitudinal (**a**) and transverse (**b**) gray-scale B mode US of the testis. Images show a well-demarcated intra-testicular lesion (blue arrow) with a small central anechoic part and a peripheral hyperechoic rim, giving an “onion-ring appearance” (white arrows). This lesion was incidentally discovered on scrotal US for a contralateral undescended testis. Serum levels of tumor markers were normal. The patient underwent conservative surgery. US, ultrasonography.

**Figure 3 cancers-14-03180-f003:**
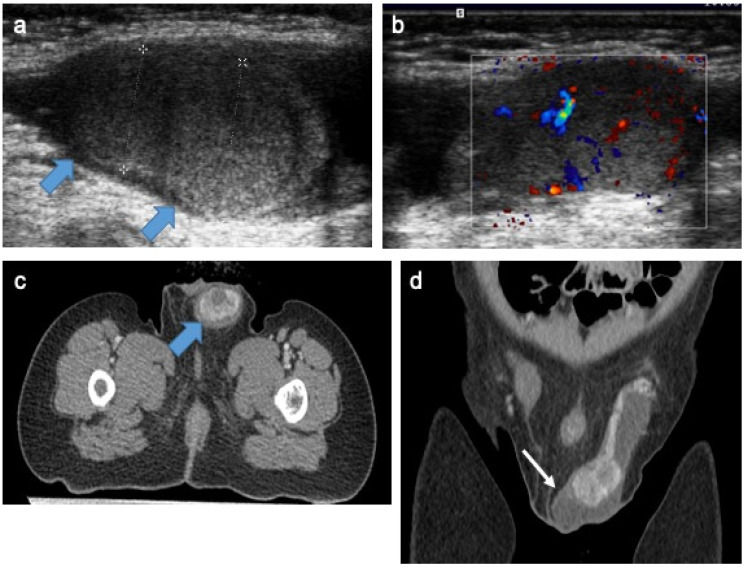
Intra-testicular yolk-sac tumor in a 20-month-old boy. Longitudinal gray-scale B mode (**a**) and transverse color Doppler (**b**) US of the testis. Staging contrast-enhanced CT focused on the pelvic on axial (**c**) and coronal (**d**) views. Images demonstrating two solid intra-testicular masses with color Doppler signal and heterogeneous enhancement after contrast (blue arrow), surrounded by peritesticular effusion (white arrow). AFP level was high and inguinal orchidectomy was performed. US, ultrasonography; AFP, alpha-fetoprotein. Courtesy of Dr Brisse and Dr Cardoen, Institut Curie, Paris, France.

**Figure 4 cancers-14-03180-f004:**
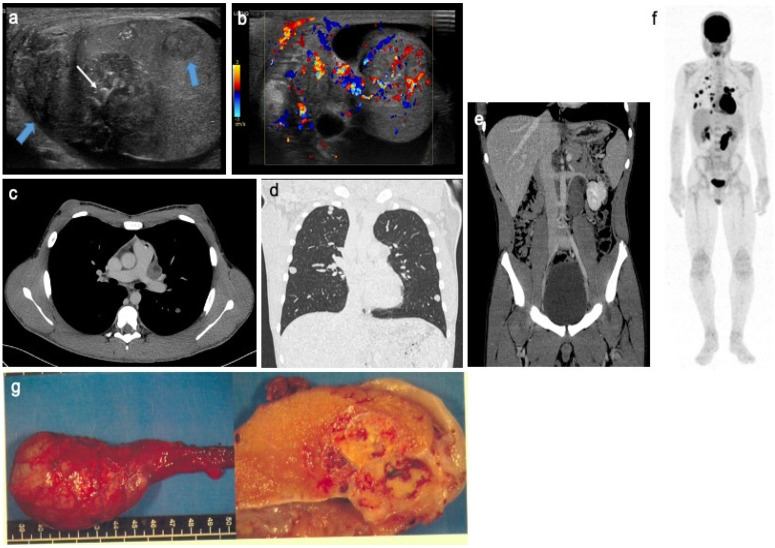
Mixed non-seminomatous malignant germ cell tumor in a 14-year-old boy with yolk-sac and choriocarcinoma components. Longitudinal gray-scale B mode (**a**) and transverse color Doppler (**b**) US of the testis. Contrast-enhanced CT on axial view of the mediastinum (**c**), coronal view of the lung (**d**) and coronal view of the abdomen and pelvis (**e**). Coronal view of ^18^F-FDG PET (**f**). Images show several heterogeneous testicular lesions (blue arrow) containing micro calcifications (white arrow) and highly vascularized tissular portions with irregular distribution (**b**). CT scan demonstrates metastatic mediastinal and retroperitoneal lymph nodes and multiple lung metastases with high uptake on ^18^F-FDG PET. (**g**) Macroscopic view with gross pathology section of the lesion after radical surgery. This adolescent presented a rapid increase in testicular volume and increased AFP and BHCG levels. US, ultrasonography; AFP, alpha-fetoprotein; BHCG, beta-human chorionic gonadotrophin; ^18^F-FDG PET, ^18^F fluorodeoxyglucose positron emission tomography. Courtesy of Dr Brisse and Dr Cardoen, Institut Curie, Paris, France and Pr Berrebi, Robert Debré Hospital, Paris, France.

**Figure 5 cancers-14-03180-f005:**
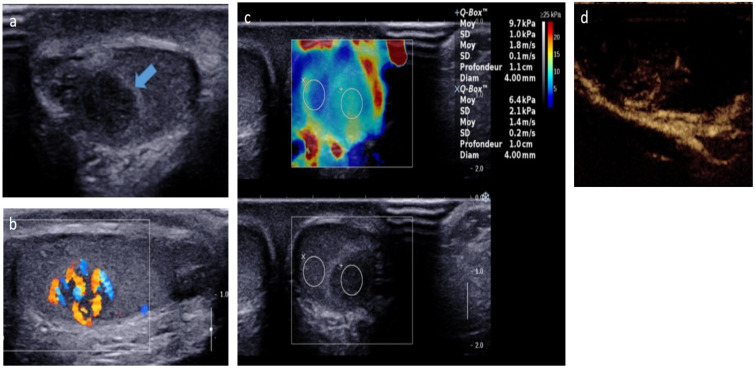
Leydig-cell tumor in an 8-year-old boy. Transverse gray-scale B-mode (**a**) and longitudinal color Doppler (**b**) US of the testis. Shear wave elastography cartography (**c**) and contrast-enhanced US focused on the lesion (**d**). Images show a well-circumscribed hypoechoic mass with intrinsic central and peripheral hypervascularization, surrounded by a hyperechoic rim (blue arrow) with moderate increased stiffness on elastography. The patient was referred for early pubic hairiness and accelerated growth rate. Testosterone level was increased, and gonadotrophin levels (follicle-stimulating hormone, luteinizing hormone) were low. The patient underwent conservative surgery. US, ultrasonography. Courtesy of Pr Laurence Rocher, Antoine Béclère, Clamart, France.

**Figure 6 cancers-14-03180-f006:**
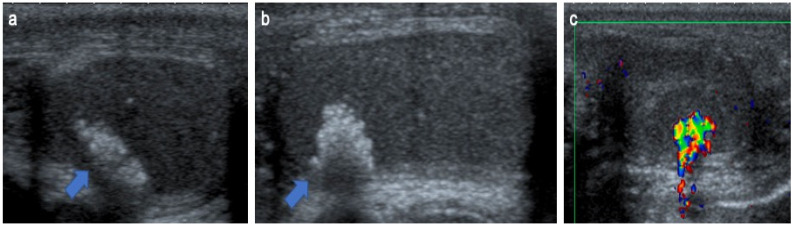
Sertoli-cell tumor in an 8-years-old boy with Peutz–Jeghers syndrome. Transverse (**a**), longitudinal (**b**) gray-scale B-mode and transverse color Doppler (**c**) US of the testis. Images show a hypoechoic nodule mostly calcified with irregular calcifications (blue arrow) causing posterior shadowing and artifacts on color Doppler imaging. US, ultrasonography. The child presented with acceleration of his growth and gynecomastia. Later, he developed nasal polyps and mucocutaneous pigmentation and Peutz–Jeghers syndrome was confirmed. The patient underwent conservative surgery.

**Figure 7 cancers-14-03180-f007:**
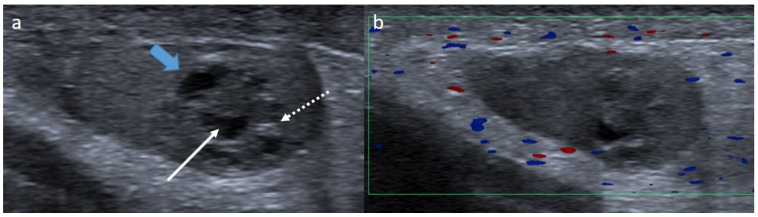
Granulosa-cell tumor in a 1-month-old boy. Longitudinal gray-scale B mode (**a**) and color Doppler (**b**) US of the testis. Images show a well-circumscribed mass with lobulated margins (blue arrow) containing a cystic component (arrow) and calcifications (dotted arrow). Serum levels of tumor markers were normal, and partial orchidectomy with an inguinal approach was performed. US, ultrasonography.

**Figure 8 cancers-14-03180-f008:**
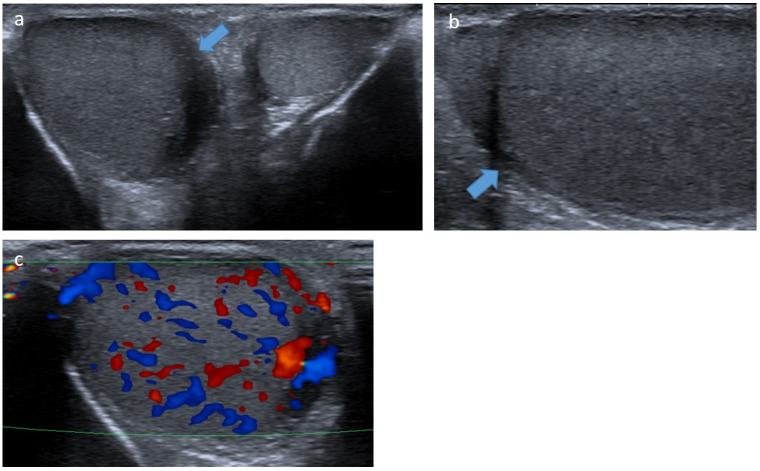
Leukemic testicular infiltration of the right testis in a 10-year-old boy. Transverse (**a**) and longitudinal (**b**) gray-scale B-mode and transverse color Doppler (**c**) US of the testis. Images show an enlargement of the right testis demonstrating an ill-defined hypoechoic infiltration (blue arrow) with increased intralesional flow preserving the normal vascular architecture. This infiltration regressed on chemotherapy. US, ultrasonography.

**Figure 9 cancers-14-03180-f009:**
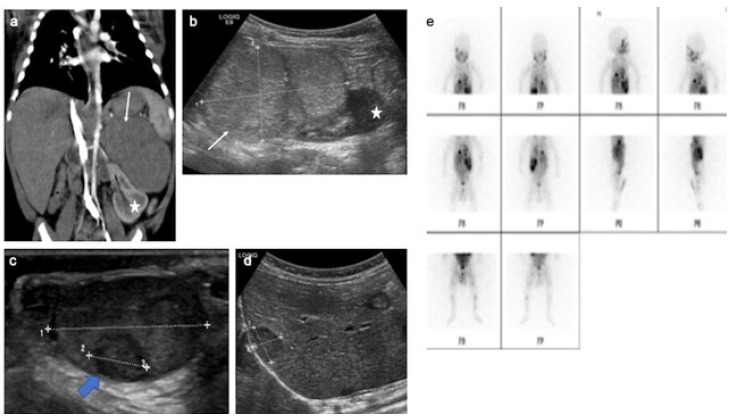
Metastatic neuroblastoma with liver, bone and testis involvement. Contrast-enhanced abdominal CT scan (**a**), different views of gray-scale B-mode US focused on the abdomen (**b**), testis (**c**) and liver (**d**). MIBG scan (**e**). Images show a left adrenal tissular mass (white arrow) with a mass effect on the left kidney (star) and intra-testicular (blue arrow) and liver metastases. The testicular metastasis appears as a well-circumscribed hypoechoic and heterogeneous mass. In addition, MIBG scan demonstrates bone extension. US, ultrasonography; MIBG, meta-iodobenzylguanidine. Courtesy of Dr Brisse, Institut Curie, Paris, France.

**Figure 10 cancers-14-03180-f010:**
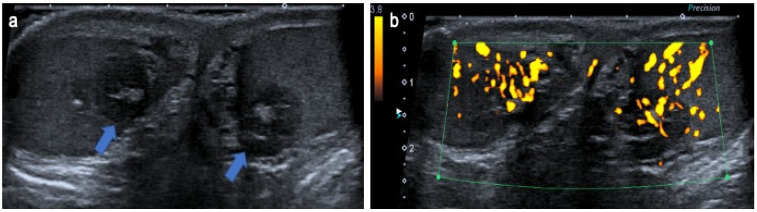
Testicular adrenal rest tumors in a 6-month boy with congenital adrenal hyperplasia. Transverse gray-scale B-mode (**a**) and color Doppler (**b**) US of the testis. Images show bilateral and symmetric lesions at the hilum of the testis, mostly hypoechoic with a hyperechoic center, lobulated margins (blue arrow) and increased intralesional flow preserving the normal vascular architecture. US, ultrasonography. Courtesy of Pr Laurence Rocher, Antoine Béclère, Clamart, France.

**Figure 11 cancers-14-03180-f011:**
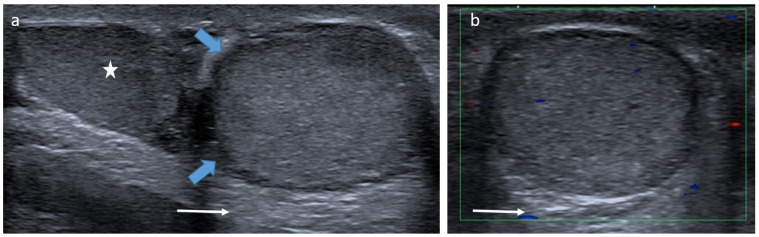
Para-testicular dermoid cyst in a 6-year-old boy. Longitudinal (**a**) gray-scale B-mode and transverse color Doppler (**b**) US of the scrotum. Images show an extra-testicular lesion (blue arrow) at the lower pole of the testis (star), demonstrating well-circumscribed margins, homogeneous echoic content with acoustic posterior reinforcement (arrow) without internal vessels on color Doppler images. Serum levels of tumor markers were normal, and the lesion was removed by an inguinal approach. US, ultrasonography.

**Figure 12 cancers-14-03180-f012:**
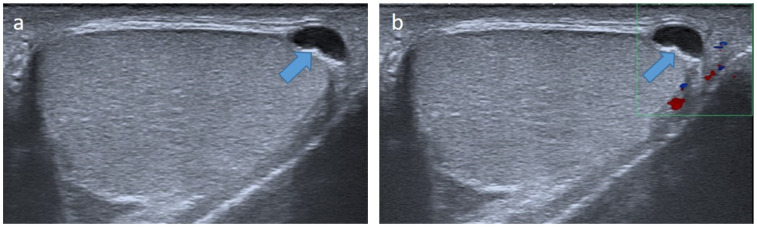
Para-testicular epithelial cyst in a 15-year-old boy. Longitudinal gray-scale B-mode (**a**) and color Doppler (**b**) US of the testis. Images show a small para-testicular cyst (blue arrow) in contact with albuginea. The lesion was removed with a scrotal approach, and pathology revealed a para-testicular epithelial cyst. US, ultrasonography.

**Figure 13 cancers-14-03180-f013:**
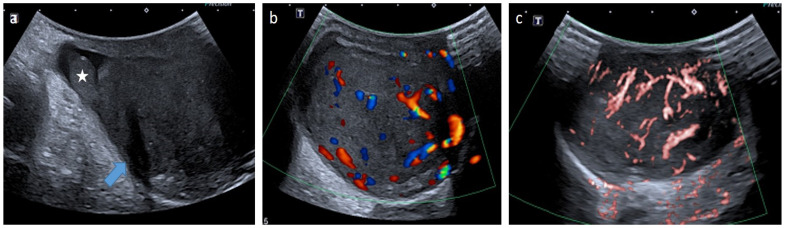
Para-testicular rhabdomyosarcoma in an 8-year-old boy (case 1). Transverse gray-scale B-mode (**a**), color Doppler (**b**) and power Doppler (**c**) US of the scrotum. Images show a solid lesion (blue arrow) in an extra-testicular location (testis is represented by a star), with heterogeneous aspect and showing high and irregular vascularization. Inguinal radical orchidectomy was performed, followed by a course of chemotherapy. The margins were safe. US, ultrasonography.

**Figure 14 cancers-14-03180-f014:**
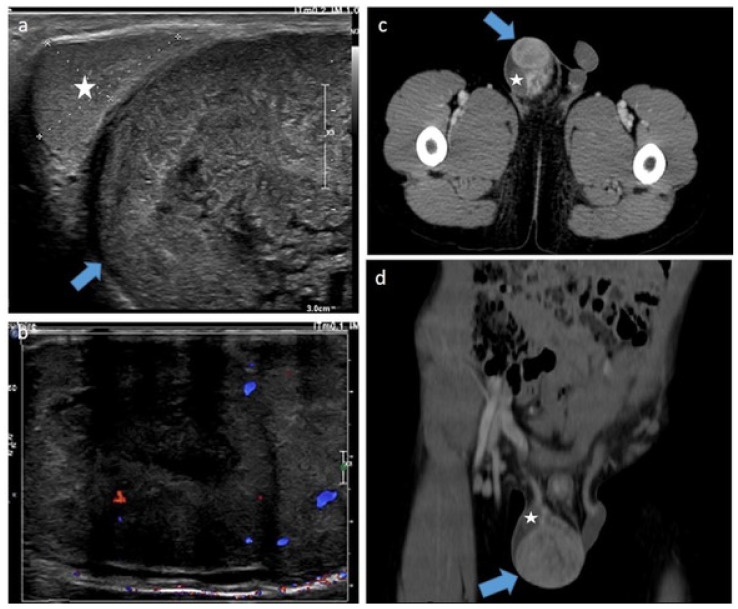
Para-testicular rhabdomyosarcoma in a 6-year-old boy (case 2). Transverse gray-scale B-mode (**a**) and color Doppler (**b**) US of the scrotum. Staging contrast-enhanced CT scan focused on the pelvic on axial (**c**) and coronal (**d**) views. Images show a solid heterogeneous lesion (blue lesion) in an extra-testicular location (testis is represented by a star) with internal vessels on color Doppler images and showing high and heterogeneous enhancement after contrast administration. US, ultrasonography.

**Figure 15 cancers-14-03180-f015:**
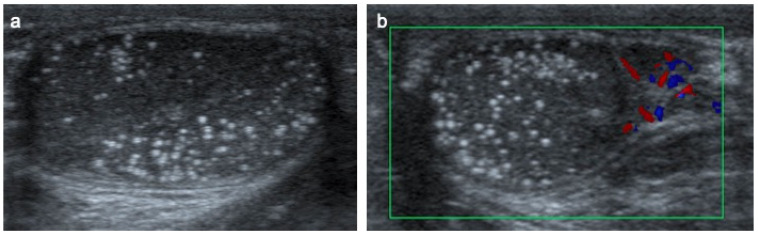
Testicular microlithiasis in a 6-year-old boy. Transverse gray-scale B-mode (**a**) and color Doppler (**b**) US of the scrotum. Images show diffuse hyperechoic punctate calcifications.

**Table 1 cancers-14-03180-t001:** Classification of intratesticular tumors in children.

Tumoral Group	Histotypes	Prevalence (%)
Benign Germ-cell tumor	Mature teratoma and dermoid cysts	40–50%
Non seminomatous malignant germ-cell tumor	Immature teratoma	rare
	Yolk sac tumor	8–30% (<2 years)
	Choriocarcinoma	rare
	Embryonal carcinoma	rare
	Mixed non seminomatous malignant germ-cell tumor	More frequent in post-pubertal patients
Seminomatous malignant germ-cell tumor	Seminoma	Exceptional in children
Sex-cord stromal tumors	Leydig-cell tumor	5%
	Sertoli-cell tumor	<1%
	Juvenile granulosa cell tumor	<5%
	Mixed sex-cord stromal tumor	<5%
Pseudo-tumors	Epidermoid cyst	10–15%
	Testicular adrenal rest tumor	30–50% in the context of congenital adrenal hyperplasia
Secondary testicular involvement of lymphoma or leukemia		1–7%
Secondary involvement of primary solid tumor		<1%
Others		<1%

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
