# Peer review of "Imaging of Pediatric Testicular and Para-Testicular Tumors: A Pictural Review"

_cancers, 2022, doi:10.3390/cancers14133180_

Round 1
Reviewer 1 Report
Well written manuscript.
Informing images and an interesting focus of this paper.
Well done.
I have only very few comments
Consider including a “pictural review” or review in the title.
Consider including a few comments on your method. How did you select the papers? Maybe a figure/table including all the tumors and % prevalence?
There seems to be a couple of errors in the list of references e.g., Ref 43 and 53.
Consider elaborating on the use of SWE elastography. There are many interesting papers on this subject, and it is important as this clinical tool can help assess testicular tumors.
What about the combination of US elastography and MRI in testicular tumors? Here is also a couple of papers on this comparing normal testis tissue with tumors and microlithiasis.
In pediatric patients, testicular microlithiasis has been reported to vanish over time. There are elastography papers on patients with microlithiasis. Consider including this information.
any thoughts on limitations?
Reviewer 2 Report
This is a well written paper on the role of different imaging modules in the diagnosis, and follow-up of testicular cancers in childhood.
The authors give a nice summary on the disease itself, which overall might give some distraction.
As a reader, and treating testicular cancer (mostly in a more elderly population), I would like to know what are the sensitivity and specificity of the different imaging modules for malignancy when compared to the final histology. In other words, why do the authors claim that more modern imaging modules are superior to the older ones if no mentioning of accuracy is performed. So, could the authors state the more modern imaging tools are better and are more likely to detect malignancy?
Could the authors give an estimation in what proportion of patients, organ sparing surgery is performed by using modern imaging techniques in stead of whole organ resecting surgery by classic imaging?
